# Exploring the Intersection of Mental and Reproductive Health Among Women Living with HIV in Spain: A Qualitative Secondary Data Analysis

**DOI:** 10.3390/healthcare13020168

**Published:** 2025-01-16

**Authors:** Ariadna Huertas-Zurriaga, David Giménez-Díez, Juan M. Leyva-Moral

**Affiliations:** 1NURECARE Research Group, Institut d’Investigació i Hospital Germans Trias i Pujol (IGTP), 08916 Badalona, Spain; ahuertasz.germanstrias@gencat.cat; 2Grup de Recerca Infermera en Vulnerabilitat i Salut (GRIVIS), Nursing Department, Faculty of Medicine, Universitat Autònoma de Barcelona, 08193 Bellaterra, Spain; david.gimenez@uab.cat

**Keywords:** reproductive health, mental health, intersectional framework

## Abstract

**Background/Objectives:** Globally, girls and women make up over half of those living with Human Immunodeficiency Virus (HIV), facing unique reproductive and mental health challenges. An HIV diagnosis impacts motherhood desires and increases trauma, stigma, and depression risks. Addressing these overlapping vulnerabilities with tailored, comprehensive healthcare is essential. This study aims to qualitatively explore the complex interplay between mental health and reproductive decision making among women living with HIV (WLWH). **Methods:** A secondary data analysis approach was employed, utilizing qualitative data from a wider study. Theoretical sampling was used to recruit a sample of WLWH attending the HIV clinic at a public hospital in Badalona (Barcelona, Spain). Interviews took place in a private and quiet space between May 2019 and January 2020. To perform secondary data analysis, Braun and Clarke’s thematic analysis was used. **Results:** Twenty-six women living with HIV were interviewed, with an average age of 39.3 years. The participants had diverse nationalities and faced economic challenges. Following data analysis, four main themes were identified: Emotional Fragmentation and Reproductive Self-Perception after the HIV Diagnosis; Coping Strategies in Reproductive Decision Making; Impacting Emotional Resilience and Motherhood Decisions; and Emotions, Mental Health, and Desire for Motherhood. The themes revealed the profound emotional impact of HIV on their identity, mental health, and reproductive decision making, affecting motherhood aspirations. **Conclusions:** Women living with HIV face psychological challenges in reproductive decision making, including anxiety and stigma. The study highlights their resilience and emphasizes incorporating hope-based strategies into HIV care, advocating for integrated and mental health-focused approaches to improve support and outcomes.

## 1. Introduction

Human Immunodeficiency Virus (HIV) is the causative agent of Acquired Immune Deficiency Syndrome (AIDS), a condition that continues to be a global public health concern [1]. Globally, girls and women make up more than half of the 37.7 million people living with Human Immunodeficiency Virus (HIV). Consequently, ending Acquired Immunodeficiency Syndrome (AIDS) by 2030 requires addressing girls’ and women’s diverse roles by putting them at the center of the response [2]. In this context, the World Health Organization (WHO) declares reproductive rights to be essential human rights and emphasizes the importance of addressing the sexual and reproductive health of women living with HIV/AIDS to safeguard their well-being as well as that of their partners and children [3,4].

The perinatal period is a transformative phase for women, [5]. During this time, women face heightened risks for various mental health issues, including depression, anxiety, obsessive–compulsive disorders, and postpartum psychosis [6]. Moreover, the prevalence of perinatal depression is much higher in vulnerable populations, such as HIV-infected women. In this regard, motherhood is considered a meaningful life role for women, rooted in sociocultural expectations and a pro-natal norm [7]. However, for women living with HIV (WLWH), HIV diagnosis can be a traumatic event, so decisions surrounding pregnancy and childbearing are complex [8,9].

WLWH face unique stressors, including challenges in relationships, disclosure, and managing sexual and reproductive health. These combined stressors often heighten mental health burdens such as depression, anxiety, and trauma-related symptoms, significantly impacting their reproductive and mental well-being [10]. Thus, the dual burden of managing HIV and navigating reproductive health decisions often leads to heightened stress and mental health concerns [11,12]. Additionally, HIV-related stigma further exacerbates social isolation, mental health concerns, and adherence to essential antiretroviral therapies (ARTs), crucial for both maternal and fetal health [13]. The perinatal period brings heightened mental health risks for WLWH. During pregnancy and postpartum, they confront not only the biological and emotional changes of motherhood but also concerns regarding perinatal HIV transmission and stigma from healthcare providers [9,14]. Furthermore, WLWH experience higher rates of perinatal depression compared to their HIV-negative counterparts, which can lead to adverse health outcomes, such as decreased ART adherence and viral suppression [15,16].

Many WLWH view motherhood as a way to restore normalcy in their lives and as a means of coping with and psychologically overcoming the challenges of their diagnosis [9]. Pregnancy is perceived by WLWH as a “way to regain their sense of womanhood and sexuality” after HIV diagnosis [14]. However, despite the documented challenges that WLWH face, psychosocial aspects of clinical care are rarely addressed by providers during conversations about reproductive desires and sexual health needs [12,17].

As documented in the literature, WLWH navigate multiple intersecting vulnerabilities: first, as individuals living with HIV; second, as those at an increased risk for mental health challenges; third, as women; and fourth, for some, as members of racial and ethnic minority groups [10]. In this context, intersectionality provides a valuable framework for understanding how power comes together and overlaps, where it interlocks and intersects. Specifically in the context of reproductive justice, this lens reveals how power structures can hinder decision making [18]. For these reasons, intersectionality is analytically significant for addressing sensitive issues like reproductive health [19]. It helps synthesize findings from the literature to identify social experiences that lead to marginalization [20] and highlights the variability in positions within and between groups that contribute to health inequities [21].

Given the prevalence of stressors and mental health challenges faced by women living with HIV and the critical connection between mental health symptoms, reproductive health, and suboptimal disease management, there is a significant gap in understanding how these factors intersect. Accessible and tailored mental health interventions, along with reproductive healthcare designed to address the unique needs of this population, are urgently needed. This study aims to address this knowledge gap by qualitatively exploring the complex interplay between mental health and reproductive decision making among WLWH. By delving into their unique challenges and experiences, this research seeks to shed light on how WLWH navigate their reproductive choices while managing mental health concerns, ultimately contributing to the development of more comprehensive and effective healthcare approaches for this vulnerable group.

## 2. Materials and Methods

### 2.1. Study Design

This study employs a secondary data analysis (SDA) approach, utilizing qualitative data from a wider study conducted by Huertas et al. [14]. SDA involves analyzing pre-existing data to address new research questions, and it is particularly valuable for exploring sensitive topics in vulnerable populations without additional participant burden [22]. The primary study had a broader objective, focusing on generating a substantive theory inductively of the phenomenon of reproductive decision making in women with HIV, using a constructivist grounded theory as the methodological approach. All twenty-six transcripts from the original study, which directly reported women’s experiences of mental health related to their reproductive experiences, were included in this new analysis.

### 2.2. Study Setting and Study Population

WLWH attending the HIV clinic at a public hospital in Badalona (Barcelona, Spain) were invited to participate. The eligibility criteria included women living with HIV who were 18 years or older, either of Spanish nationality or residing in Spain for at least five years, and fluent in Spanish or Catalan. Women who were institutionalized in prisons or psychiatric centers, as well as those with fragile health conditions, were excluded from the interviews. A detailed account of recruitment and sampling can be found in the principal study’s methodology [14]. Theoretical sampling [23] was used to ensure that the participants selected could provide rich and relevant data to address the research questions. Participants were chosen based on factors such as their HIV status, reproductive health experiences, and sociodemographic factors, with the goal of capturing a diverse range of perspectives. This approach allowed for the inclusion of participants with varying experiences, ensuring that theoretical saturation was reached and that the findings were robust and reflective of the broader population’s experiences. The focus of the study has been deliberately maintained on Spain to ensure alignment with the sociocultural context, allowing for the collection of realistic and contextually relevant data. This approach enhances the study’s rigor by grounding the findings within a specific and consistent cultural and social framework, as recommended in qualitative research to ensure credibility and transferability. The final sample size was unknown until the study progressed and theoretical data saturation was achieved.

### 2.3. Study Instruments

Semi-structured interviews and a clinical and sociodemographic questionnaire were used as data collection instruments. The inclusion of these clinical and sociodemographic characteristics primarily aimed to provide a clearer understanding of the sample, enhancing the transferability and credibility of the findings. The interviews served as the primary tool to address the main objective of the study. Regarding the interviewing, a detailed account of the process can be found in the principal study [14]. The interviews took place in a private and quiet space outside the HIV clinic unit between May 2019 and January 2020. They ranged from 8 to 58 min in length (mean = 24 min). The interviews were transcribed verbatim post-interview; for this SDA, anonymized transcripts were stored on a password-protected, encrypted folder on a secure university server. The transcripts were then uploaded to NVivo, where they were stored as a password-protected project.

### 2.4. Ethical Aspects

The data were collected once the Ethics Committee at Hospital Universitari Germans Trias i Pujol approved the study (reference number: PI-18-141). Informed consent was obtained from all subjects. Participation was voluntary and anonymous, and no identifying information was recorded. The participants’ real names were changed to nicknames for anonymity. The participants were able to withdraw at any time during the research without penalty.

### 2.5. Analysis

To perform SDA, Braun and Clarke’s thematic analysis was employed, as it assists researchers in systematically examining perspectives [24]. All transcripts involved in the original study were read to confirm which were relevant to the SDA. The transcripts were re-read to assist familiarization and reflection on potential codes. NVivo was used to help manage the large volume of transcripts.

Analysis was conducted by three experienced postdoctoral researchers specialized in nursing, vulnerability, and mental health (AH, JL, and DG). A list of codes, derived from the data, was created to simplify the reviewing process. A thematic mind map helped to visualize the relationships between codes and arrange them into potential themes; this step was repeated following discussions within the research team until the best fit was achieved. The final list of themes is summarized in Table 1.

### 2.6. Measures of Rigor

To ensure the trustworthiness of this secondary data analysis, measures of rigor were implemented throughout the analytical process. Reflexivity and critical thinking guided the entire analysis, with regular discussion meetings held between the three researchers to verify and agree upon the findings. This approach aligns with established practices for enhancing the credibility and dependability of qualitative research [25]. The researchers engaged in continuously refining themes and interpretations based on team discussions. This collaborative approach to analysis helped to mitigate individual biases and enhance the depth of interpretations, contributing to the overall rigor of the study.

## 3. Results

In the original study, twenty-six women were interviewed, enough to reach theoretical saturation. All experiences analyzed pertain to women living with HIV aged 18 to 50 years, with an average age of 39.3 years (SD: 8.97). For diversity and inclusivity, a transgender woman was also invited to participate. Sixty-five percent of the participants were from Spain, while six were from South America, two were from Africa, and one was from Ukraine. Nearly half of the interviewees reported an annual income below EUR 5000, another half earned between EUR 5000 and EUR 25,000, and only one participant earned above this range. Regarding educational levels, 53% of the informants had completed secondary education, four had higher education, six had primary education, and two had no formal education.

Regarding health, 73% of participants were diagnosed with HIV for over ten years, with sexual transmission being the most common infection route. All women had an undetectable viral load (below 50 copies/mL), and over 60% had CD4+ levels above 500 cells per cubic millimeter. Of the twenty-six women interviewed, eighteen had a partner, fourteen of whom were in serodiscordant relationships. Twenty of the participants had children, eight of whom had all their children before their HIV diagnosis.

Following data analysis, four main themes were identified:Emotional Fragmentation and Reproductive Self-Perception after the HIV Diagnosis;Coping Strategies in Reproductive Decision Making;Impacting Emotional Resilience and Motherhood Decisions;Emotions, Mental Health, and Desire for Motherhood.

The HIV diagnosis has a significant emotional impact on the affected women, affecting their identity and psychological well-being and directly influencing how they approach motherhood and reproductive decisions (Figure 1).

### 3.1. Emotional Fragmentation and Reproductive Self-Perception After the HIV Diagnosis

The emotional fragmentation experienced by women following an HIV diagnosis, marked by a fear of rejection, loss of normalcy, and self-perception as “dirtiness”, profoundly affects how they perceive their capacity to be mothers. The participants also experienced guilt, shame, fear, and other related emotions.

#### 3.1.1. Fear of Rejection and Stigmatization in Motherhood

The fear of social and familial rejection due to their HIV-positive status creates a psychological barrier that often prevents women from openly expressing their desire to become mothers.

“Because if society looks down on you, thinking you’re harming your child, right? Then, that’s a stigma you carry with you.” (Clotilde, p. 3)

Some women may avoid or delay reproductive decisions out of fear of judgment or increased discrimination within their social and family circles. This fear not only suppresses their reproductive desires but also compels them to hide their intentions to start a family.

“When I was diagnosed, I thought I was going to die, that I would never find a partner, that no man would want me because of the disease… I thought, ‘I’ll never be able to be a mother.’ And, of course, without information, you create this world, you think you can’t do anything, that no man will love you… That’s just what goes through your mind.” (Victoria, p. 12)

“I always had it in my mind that I was HIV-positive, and that often held me back from relationships. My self-esteem plummeted; I felt awful, emotionally low, thinking no one would ever take me seriously because of my condition.” (Emmeline, p. 3)

#### 3.1.2. Loss of Normalcy and Seeking Validation Through Motherhood

The feeling of having lost a sense of normalcy after the HIV diagnosis can also influence the decision to pursue motherhood as a means to regain a socially accepted identity. For some women, motherhood represents a way to reconnect with the concept of normalcy and fulfill traditional feminine roles, allowing them to reaffirm their self-worth despite emotional fragmentation: “Motherhood is something everyone desires.” (Amparo, p. 5)

### 3.2. Coping Strategies in Reproductive Decision Making

#### 3.2.1. Negotiating My Identity, Taking Care of Myself, and the Desire to Be a Mother

The coping strategies women with HIV use to manage the emotional impact of their diagnosis also influence their reproductive decisions. In their journey of identity reconstruction, many women with HIV find themselves having to reconcile their illness with their capacity to be mothers. The decision to have children often becomes part of this negotiation, as they try to strike a balance between their desire for motherhood and the reality imposed by HIV. This negotiation process is not always easy, as they are influenced by external prejudices and by the stigma that links HIV with an inability to be a “good mother”. However, motherhood becomes a coping strategy, fostering self-care (taking care of myself and others): “taking the medication for both you and him to avoid transmission risks… The important thing here is the medication.” (Griselda, p. 7)

“… because from the moment I had her, I began to take care of myself. Before, I liked to go out dancing, drinking, all that. I haven’t stopped entirely, but I do it less. [laughs] I used to stay out late at a club drinking, but not anymore. Things happen that make you slow down. If you used to live too fast and now want to live and watch your daughters grow, you have to slow down the pace of life you once had. Sometimes things happen to help you realize that and start slowing down.” (Amparo, p. 4)

#### 3.2.2. Negotiating My Sexuality

Fear of rejection and of transmitting HIV to their sexual partners force women to negotiate and re-define their sexual relationships. Even HIV leads some women to choose celibacy. This avoidance strategy regarding romantic and sexual relationships directly impacts their reproductive decisions, as it involves an implicit or explicit renunciation of motherhood. For these women, celibacy can be a way to avoid the emotional distress associated with the possibility of transmitting HIV to a child or partner or of facing a society that does not easily accept the idea of an HIV-positive woman becoming a mother.

“When you were diagnosed, you told me you wanted to be a mother. Flora: No, what happened is, well, it was impossible. First, finding a partner…” (Flora, p. 5)

“I wouldn’t feel right knowing I’m sick, transmitting it to the child, and having them always be sick. I can’t handle that. I’d rather not have one.” (María, p. 3)

### 3.3. Impacting My Emotional Resilience and Motherhood Decisions

#### 3.3.1. Lack of Emotional Support and Reproductive Decisions

Factors that promote or hinder emotional resilience are deeply intertwined with reproductive decisions. The absence of emotional support from family, partners, or friends significantly impacts reproductive decisions. Women with HIV who lack a support network may feel they do not have the strength or emotional stability necessary to face motherhood, often leading them to postpone or abandon their desire to have children. Emotional isolation becomes a major barrier to making reproductive decisions in a safe and trusting environment.

“No one here supported me, my family didn’t support me, everyone kept saying, ‘Oh, no, you’re going to get sick, you’re going to die, you won’t handle the pregnancy, your child will die…’” (Marie, p. 2)

“My life is very lonely; now I live alone, always alone, I don’t talk to anyone, and I don’t want to hear from anyone because I’ve had so many bad experiences that I’d rather stay locked in my own world, in my world, just me. And whatever problems come, let them come [you let go], because they’re coming toward me, not because anyone else is causing them.” (Dolores, p. 8)

#### 3.3.2. Stigmatization in Healthcare Settings

Women with HIV who encounter stigmatizing attitudes within the healthcare system may feel even more vulnerable when making reproductive decisions. The fear of being judged for wanting to be mothers or of not receiving adequate information and support may inhibit their reproductive desires.

“… for a blood test, they put on three pairs of gloves.” (Maya, p. 9)

“I feel it in my soul because the gynecologist here before… I don’t know, I’ve had a few, but the one I had before told me I was crazy, that I was killing myself and my child.” (Marie, p. 2)

“My gynecologist insisted I have a C-section to prevent transmission, while the other woman, who I think was the obstetrician, said no, that I could still have a natural birth. So it was a debate between them, which made me feel very uncomfortable. I thought, ‘Am I being selfish by wanting a C-section because I’m scared, or would a natural birth actually be better for him?’” (Clotilde, p. 4)

In many cases, women feel they lack sufficient control over their reproductive decisions, leading them to defer these choices to other figures, such as healthcare professionals or their partners.

“Then I went to her [the doctor], and she told me, ‘Don’t worry; there are many women, and the percentage of children born with HIV is very low.’ And so I just let go. With the first child, you just let go, you do what they tell you, you take the medication, and they tell you to stay calm.” (Maya, p. 2)

#### 3.3.3. Seeking Resources for Emotional Resilience: Support Networks, Spirituality, and Psychological Help

WLWH seek resources for their emotional resilience. Sharing their story with other women who have also become mothers is a learning experience and a way to combat the isolation that can accompany their journey into motherhood. Additionally, women describe very positive experiences when they seek psychological support to adapt to their diagnosis, disclose their HIV status, resolve questions, or share concerns, including the dilemma of motherhood. Other strategies women with HIV use include drawing strength from spiritual beliefs, which provide them with the resilience to carry on, and sometimes, they feel everything is already planned out for them.

“I was on an HIV-positive page, and I started to post, and yes, a lot of mothers wrote to me, telling me to stay calm. That actually gave me peace of mind.” (Maya, p. 4)

“Whenever I have a question, I come, and I ask her, and she gives me advice… Talking to the psychologist always relaxes me, gives me advice, and reassures me that nothing will happen to me… and that calms me down.” (Juana, p. 3)

### 3.4. Emotions, Mental Health, and the Desire for Motherhood

#### 3.4.1. Anxiety and Persistent Fear

The prolonged impact of HIV on mental health also affects expectations regarding motherhood and reproductive decisions. Despite the very low risk of HIV transmission during pregnancy with appropriate treatment, women with HIV continue to experience anxiety about the possibility of passing the virus to their children. This fear, compounded by concerns about the potential effects of treatment on the fetus, makes motherhood feel like a high-stakes emotional risk. The ongoing uncertainty affects their ability to make informed and secure reproductive choices.

“Above all, you see, I’ve always been very afraid of passing anything on to him, so when it comes to reproduction, passing something on to him, or even during childbirth with my daughter, I was scared of passing something on to the baby.” (Victoria, p. 1)

“You’re always afraid. Even though there are so many advances, even if everything’s well managed… It’s true that today medications change quickly, and you protect the person, you protect the baby. But you’re afraid. You always think something might go wrong.” (Clotilde, p. 3)

“…when I started with him, I was afraid that… Because they always tell you, my doctor used to tell me that there’s a certain % risk. So, I always told him: ‘What if you end up being that %?’ I’ve always tried to prevent it because I just don’t want that for him.” (Clara, p. 3)

#### 3.4.2. Navigating Uncertainty, Depression, and Hope in the Journey to Motherhood

In some cases, depression resulting from HIV and uncertainty may lead women to lose hope of becoming mothers. The feeling that their life is limited by illness, combined with the perception that they cannot fulfill traditional caregiving roles, can diminish their desire to have children. This sense of hopelessness becomes an emotional barrier limiting reproductive options:

“Because you don’t know what could happen. Not just your health, but you don’t know what could happen tomorrow. If you have a healthy child who can fend for themselves, who can go through life on their own, you’re more at peace because anyone could take care of your child. But if you have a sick child, people hesitate more about whether to care for them if you’re not around.” (Simone, p. 2)

In contrast to the negative effects on mental health, hope also emerges as a key factor promoting the desire for motherhood. For many women, motherhood and raising healthy children become a source of hope and strength that motivates them to continue fighting against HIV. Motherhood represents an opportunity to overcome the emotional and physical challenges of HIV, providing them with meaningful purpose in life: “…I have to fight [for my children]; I won’t let this disease defeat me.” (Emmeline, p. 2)

“What were your reasons to keep going?” Simone: “My daughters. My daughters were my reason; I had a very young daughter, and I couldn’t allow myself to leave her alone. That was my main reason to say I have to take care of myself.” (Simone, p. 2)

HIV has a significant impact on the mental health of women, directly affecting their reproductive decisions. Emotional fragmentation following diagnosis, the fear of rejection and stigma, and coping strategies shape how these women perceive and face reproductive decisions. Although many of them experience emotional barriers that complicate their reproductive choices, the desire to be a mother often remains, serving as a means to reconnect with their identity and create hope in their lives. The relationship between mental health and reproduction in women with HIV is complex and multifaceted, highlighting the need for a comprehensive approach to emotional and psychological support for these women.

The reproductive decisions of women living with HIV (WLWH) are shaped by a complex interplay of psychological, social, and emotional factors. The fear of social and familial rejection creates psychological barriers to openly expressing their desire for motherhood, while the diagnosis itself often disrupts their sense of normalcy, pushing some women to see motherhood as a way to reclaim their identity and self-worth. Coping strategies, including identity reconstruction and balancing the desire for motherhood with the realities of living with HIV, illustrate how reproductive decisions are deeply tied to emotional resilience and support systems. External factors, such as stigma from society and the healthcare system, amplify emotional isolation and hinder decision making. Simultaneously, internal challenges, like anxiety over transmission risks and depression, further complicate the ability to make informed and secure reproductive choices. However, despite these barriers, hope and resilience often emerge as powerful motivators. For many women, motherhood represents a source of strength, a coping strategy, and a way to create meaning in their lives, underscoring the importance of comprehensive psychological and emotional support for WLWH navigating these decisions.

## 4. Discussion

The findings of this study provide insights into the complex inter-relationship between HIV, mental health, and reproductive decision making among women living with HIV (WLWH). Previous research has emphasized the disruptive impact of an HIV diagnosis on women’s emotional well-being and sense of identity, with long-term consequences for their psychological health and life plans [26,27]. In addition, this study reveals how emotional and psychological stressors can have a significant impact on reproductive choices. A substantial number of WLWH experience considerable anxiety and depression when considering motherhood, predominantly driven by concerns about vertical transmission and worries about their capacity to maintain their health while raising a child [28,29]. This sustained emotional burden can result in a state of decision paralysis, characterized by a lack of commitment to a definitive reproductive path due to a sense of internal conflict between the desire for motherhood and the perceived risks involved [30].

The present study indicates that stigma has a pervasive impact on reproductive decision making. Both external stigma, such as discrimination within healthcare settings and social exclusion, and internalized stigma, which manifests as guilt, self-doubt, and shame, create significant obstacles for WLWH [8,31]. These additional layers of stigma serve to further complicate an already challenging emotional landscape. Despite their HIV status, women are frequently compelled to demonstrate their capacity and desirability as mothers [32,33,34]. This dichotomy exemplifies the conundrum of navigating two conflicting identities: that of managing a chronic illness and that of aspiring to become nurturing and resilient mothers who can provide love, stability, and security [35]. To effectively address the overlapping identities of these women requires the implementation of holistic care strategies that acknowledge not only the clinical challenges but also the profound emotional difficulties they face. Moreover, this study contributes to the existing body of knowledge by examining the role of social isolation and the provision of emotional support in shaping reproductive decisions. While Ji et al. [36] have previously documented the negative impact of isolation on WLWH, our findings specifically connect this isolation to the reproductive sphere. In the absence of adequate support networks, whether from partners, family, friends, or compassionate healthcare providers, many women feel ill-prepared to confront the uncertainties and vulnerabilities associated with considering motherhood [37]. This lack of emotional support often exacerbates stress, leading WLWH to retreat into ambivalence or avoidance, which ultimately undermines their ability to make confident, informed choices.

The present study also serves to reinforce earlier observations that have identified fears of HIV transmission as a significant source of anxiety, notwithstanding the medical advancements that have led to a notable reduction in the likelihood of vertical transmission [26]. These findings demonstrate the inadequacy of relying on clinical solutions alone. It is evident that a more integrated approach is required, one that addresses the emotional concerns and persistent fears that influence decision making. This is particularly pertinent when considering the long-term mental health trajectories associated with different reproductive choices. For example, Moseholm et al. [29] discovered that while the decision to become a mother can initially elevate anxiety levels, it frequently results in enhanced mental health outcomes over time. Conversely, women who remain undecided or choose not to pursue motherhood may experience prolonged emotional distress. The recognition of these distinct trajectories highlights the necessity for the provision of longitudinal, adaptable mental health support that is responsive to the evolving circumstances and decisions of women.

The integration of routine mental health screenings into HIV care represents a pivotal initial measure in addressing these concerns [27,38]. The early identification of anxiety, depression, or trauma-related symptoms enables the implementation of timely interventions, thereby preventing the establishment of these challenges as deeply entrenched issues. Furthermore, the implementation of trauma-informed care practices [26] guarantees that healthcare interactions are empathetic, respectful, and responsive to past experiences of discrimination or abuse. Another essential element is specialized reproductive counselling [32], which provides WLWH with accurate information about the prevention of mother-to-child transmission (PMTCT) while also acknowledging the emotional implications of these decisions. Cognitive behavioral strategies [39] can serve to further empower women by providing them with the tools to effectively manage anxiety and build confidence in navigating the complexities of reproductive choices. Peer support networks are similarly beneficial, as they provide a forum for WLWH to discuss their concerns, challenges, and aspirations while also fostering a sense of community and validation [10].

The principal finding of this study is that integrated, patient-centered care is a fundamental element in the provision of optimal healthcare. It is imperative that healthcare providers move beyond the limitations of strict clinical approaches and acknowledge the complex interconnections between stigma, mental health, and identity formation and their intersections with reproductive intentions. By fostering open and judgment-free dialogue and delivering empathetic care that is tailored to the individual, healthcare providers can create an environment in which women living with HIV (WLWH) feel respected, understood, and supported. Such an approach would empower women to make reproductive choices that align with their values, goals, and well-being. While this study reaffirms existing evidence on depression, anxiety, stigma, and persistent concerns about transmission among women living with HIV (WLWH), it also offers a more nuanced understanding of the underlying mechanisms driving these patterns.

### 4.1. Implications for Practice

It is recommended that healthcare providers implement comprehensive, integrated care models that address the multifaceted needs of WLWH. This should include routine screening for mental health issues, particularly anxiety and depression, as part of regular HIV care. Providers should be trained in trauma-informed care approaches to create a supportive, non-judgmental environment that acknowledges the impact of stigma and past negative experiences on women’s health-seeking behaviors and decision-making processes. Additionally, reproductive health counselling for WLWH should be enhanced to provide accurate, up-to-date information on the prevention of vertical transmission and other issues related to pregnancy and HIV, such as addressing the emotional and psychological aspects of reproductive decision making, helping the women navigate complex feelings and societal pressures. Healthcare providers should be prepared to offer ongoing support and guidance, as the women’s reproductive intentions may change over time.

Mental health support should be integrated into HIV care services, with access to psychological interventions such as cognitive behavioral therapy. These interventions should be tailored to address the specific concerns of WLWH, including managing stigma, building self-esteem, and developing coping strategies for anxiety related to disclosure and reproductive choices. Healthcare systems should prioritize the development of peer support networks for WLWH. These networks can provide valuable emotional support, reduce social isolation, and offer practical advice from women with shared experiences. Furthermore, healthcare systems should prioritize the reduction of stigma within healthcare settings through staff education and the implementation of anti-discrimination policies, with all staff, from receptionists to specialists, being trained in the provision of respectful, confidential care to WLWH.

### 4.2. Future Directions

Future directions for research on WLWH should concentrate on several key areas to address the unique challenges and needs of this population. Firstly, further research is required on integrated care models that combine HIV treatment with sexual and reproductive health services, mental health support, and the management of other chronic conditions, such as hypertension and diabetes. Studies should evaluate different strategies for integration across various health systems and social contexts, including approaches that can improve uptake of services, such as contraception and cervical cancer screening. Secondly, longitudinal research is required to examine the long-term health outcomes of WLWH, particularly as they age and face increased risks of conditions like cardiovascular disease. This should include an assessment of the effectiveness of interventions to promote healthy lifestyles and manage comorbidities. Thirdly, further work is necessary to develop and evaluate trauma-informed, culturally appropriate mental health interventions for WLWH. This should include examining the cost effectiveness of different psychosocial support models and identifying strategies to overcome barriers to mental healthcare access and utilization. Additionally, research should focus on improving engagement and retention in care across the HIV treatment cascade. This includes developing tailored approaches for tracing and re-engaging WLWH who have fallen out of care, as well as identifying accurate and feasible measures of treatment adherence. Finally, the implementation of research is crucial to understand how to effectively scale up evidence-based interventions for WLWH in real-world settings. This should involve community-based participatory approaches that meaningfully involve WLWH in the research process from conceptualization through dissemination.

### 4.3. Limitations

This study presents some limitations that must be considered. While SDA offers numerous advantages, such as cost effectiveness and access to large datasets, it also presents limitations that should be addressed [40]. As noted by Ruggiano and Perry (2019), SDA may limit researchers’ ability to fully explore certain aspects of the phenomenon due to constraints in the existing dataset. Also, the data were not collected specifically for our research questions, which may result in some variables being unavailable or not ideally suited to our study aims. Moreover, our analysis is inherently limited by the inability to definitively examine causality due to its retrospective nature [40]. Additionally, the potential for researcher bias in secondary analysis should be acknowledged, as prior knowledge of the data may influence the analytical approach [41]. The temporal relevance of the data must also be considered, as the original interviews were conducted between May 2019 and January 2020, potentially not reflecting the most current realities of women living with HIV, especially given the rapidly evolving healthcare landscape. Finally, the study’s focus on a specific geographical area (Barcelona, Spain) may limit the generalizability of findings to other cultural contexts.

## 5. Conclusions

This study reveals that WLWH face significant psychological challenges when making reproductive decisions, including anxiety about vertical transmission, concerns about their long-term health, and struggles with stigma and identity. However, it also highlights the resilience and adaptive strategies employed by these women, with hope emerging as a key factor in their decision-making process.

Recognizing hope as a crucial element in reproductive decisions suggests opportunities for strength-based interventions that are currently underutilized in HIV care. By incorporating hope-focused strategies into care plans, healthcare providers may be able to better support WLWH in their reproductive decision-making process while simultaneously addressing their mental health needs.

The research underscores the need for integrated care models that address both the physical and psychological aspects of HIV management and reproductive health. It emphasizes the importance of routine mental health screenings, trauma-informed care, specialized reproductive counseling, and peer support facilitation in providing comprehensive care for WLWH.

Future research directions should focus on developing and evaluating interventions that specifically address the psychological aspects of reproductive decision making among WLWH. Additionally, investigating the long-term effects of hope-focused interventions on mental health and reproductive outcomes could offer critical insights for improving care for this population.

## Figures and Tables

**Figure 1 healthcare-13-00168-f001:**
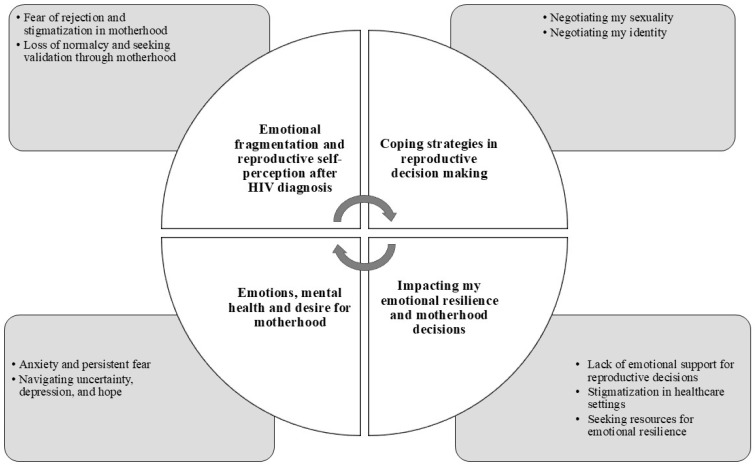
Representation of the intersection of mental and reproductive health among WLWH.

**Table 1 healthcare-13-00168-t001:** Themes, subthemes, and complex codes.

Theme 1	Emotional Fragmentation and Reproductive Self-Perception After the HIV Diagnosis		
Subthemes	Fear of rejection and stigmatization in motherhood	- Loss of normalcy and seeking validation through motherhood	
Complex codes included	- Self-stigmatization - Discrimination and rejection - Prejudice and discrimination within the healthcare community - Complexity of disclosing serostatus	- Unfulfilled reproductive desires, broken dreams - Motherhood is everything - Motherhood is life’s law - Motherhood as a path to normalcy	
Theme 2	Coping Strategies in Reproductive Decision Making		
Subthemes	Negotiating my identity, taking care of myself, and the desire to be a mother	Negotiating my sexuality	
Complex codes included	- My identity as a person living with HIV and as a mother - Taking care of myself and others - Adherence to treatment, trust in its safety, efficacy, and physical and emotional well-being - When you want to, you care; partner protection - Child protection, “I want my children to be healthy”	- Celibacy as a protection strategy - Negotiating safe sexual relationships	
Theme 3	Impacting Emotional Resilience and Motherhood Decisions		
Subthemes	Lack of emotional support and reproductive decisions	Stigmatization in healthcare settings	- Seeking resources for emotional resilience
Complex codes included	- “They don’t even know I have HIV”; lack of family support - I can’t count on my partner	- Lack of support from professionals - They have never asked me if I wanted to be a mother - My rights were violated when I was sterilized; imposed decision	- Empowering myself - Support networksexperiences of other women with HIV - Peer relationshipsPsychological support - Spiritual forces
Theme 4	Emotions, Mental Health, and Desire for Motherhood		
Subthemes	Anxiety and persistent fear	Navigating uncertainty, depression, and hope in the journey to motherhood	
Complex codes included	- False myths and beliefs about HIV and motherhood - Fear of horizontal and vertical transmissionFear of the possible side effects of antiretrovirals on the fetus - Fear of not being able to care for my children and leaving them alone - Irrational fears	- Constant suffering: HIV is always there - Challenging myself: fears, uncertainties, and doubts - Motherhood as hope and a reason to keep fighting against HIV - Recovering lost opportunities for motherhood	

## Data Availability

The raw data supporting the conclusions of this article will be made available by the authors on request.

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
