# Peer review of "Exploring the Intersection of Mental and Reproductive Health Among Women Living with HIV in Spain: A Qualitative Secondary Data Analysis"

_healthcare, 2025, doi:10.3390/healthcare13020168_

Round 1
Reviewer 1 Report
Comments and Suggestions for Authors
A review of the manuscript entitled “Exploring the intersection of Mental and Reproductive Health Among Women Living with HIV in Spain: A Qualitative Secondary Data Analysis”
1. (page 1, line 18): Please give the meaning of SDA acronym on the first mention.
2. Authors need to confirm that all acronyms are defined before being used for the first time. For example, HIV and AIDS.
3. I recommend that the authors include the study by Djojosugito et al. titled "Prevalence of major INSTI and HIV-1 drug resistance mutations in pre- and antiretroviral-treated patients in Indonesia" in the introduction. This study provides a clear definition of HIV and AIDS, which could significantly enhance the quality of this manuscript. Thank you!
4. (page 2, lines 44-47): This sentence “…motherhood is considered a meaningful life role for women, rooted in sociocultural expectations and a pro-natal norm” will be more significant if it’s also supported by another relevant study. Please check https://doi.org/10.52225/narra.v3i3.405
5. Authors are suggested to proofread the manuscript after addressing all comments to avoid any typological, grammatical, and lingual mistakes and errors. For example, “orignal” on page 3 line 108; “min” on page 4 line 122; and “Negotiating My Identity, taking care of myself and the Desire to Be a Mother” on page 5 table 2.
6. (page 3, 104-105): What distinguishes this study from Huertas et al.'s? How does the research question differ from the previous study?
7. (page 3, lines 112-116): What were the inclusion and exclusion criteria for the study?
8. (page 9, discussion): The findings described by the author correlate well with the results. However, important limitations related to use of Secondary Data Analysis should be highlighted.
Comments on the Quality of English LanguageThe quality of English language is good and understandable.
Author Response
- (page 1, line 18): Please give the meaning of SDA acronym on the first mention.
- We apologize. The meaning of SDA acronym has been provided.
- Authors need to confirm that all acronyms are defined before being used for the first time. For example, HIV and AIDS.
- All the acronyms used in the paper have been revised and defined.
- I recommend that the authors include the study by Djojosugito et al. titled "Prevalence of major INSTI and HIV-1 drug resistance mutations in pre- and antiretroviral-treated patients in Indonesia" in the introduction. This study provides a clear definition of HIV and AIDS, which could significantly enhance the quality of this manuscript. Thank you!
- Thanks for the suggestion. The reference indicated by the reviewer has been included.
- (page 2, lines 44-47): This sentence “…motherhood is considered a meaningful life role for women, rooted in sociocultural expectations and a pro-natal norm” will be more significant if it’s also supported by another relevant study. Please check https://doi.org/10.52225/narra.v3i3.405
- We appreciate your comment. We have read the paper and we consider that this article does not address the importance of the maternal role in the lives of many women. In fact, it discusses The Impact of the Triple Elimination Program for Mother-to-Child Transmission of HIV, Syphilis, and Hepatitis B in Indonesia, and therefore it is deemed not appropriate for inclusion.
- Authors are suggested to proofread the manuscript after addressing all comments to avoid any typological, grammatical, and lingual mistakes and errors. For example, “orignal” on page 3 line 108; “min” on page 4 line 122; and “Negotiating My Identity, taking care of myself and the Desire to Be a Mother” on page 5 table 2.
- We apologize. The whole paper has been proofread to avoid all English mistakes.
- (page 3, 104-105): What distinguishes this study from Huertas et al.'s? How does the research question differ from the previous study?
- We have indicated in the revised version that the primary study had a broader objective, focusing on generating a substantive theory inductively of the phenomenon of reproductive decision-making in women with HIV, using a constructivist grounded theory as the methodological approach.
- (page 3, lines 112-116): What were the inclusion and exclusion criteria for the study? Thank you for your comment.
- We have specified that eligibility criteria included women living with HIV who were 18 years or older, either of Spanish nationality or residing in Spain for at least five years, and fluent in Spanish or Catalan. Women who were institutionalized in prisons or psychiatric centers, as well as those with fragile health conditions, were excluded from the interviews.
- (page 9, discussion): The findings described by the author correlate well with the results. However, important limitations related to use of Secondary Data Analysis should be highlighted.
- Thank you for your comment. A limitations paragraph has been included
Reviewer 2 Report
Comments and Suggestions for Authors
Thank you for the opportunity to review this well written manuscript. Authors have clearly spent time and effort to put this together. Overall, the manuscript is well written and the methods are well described. Findings are well presented and the discussion & conclusion is grounded in the findings of the study. I have a few comments for improvement.
Introduction
1. The introduction is quite long (unnecessarily). Consider tightening it by summarizing some of the literature review and focusing on the specific aims of the current study. Line 41 – 67 could be consolidated into one paragraph.
2. More emphasis could be placed on the specific gap in knowledge that this research aims to address.
3. There are some gen-AI phrasing patterns. I suspect authors used AI for language editing. Authors should disclose use of gen-AI if not already done so.
Methods
1. The methods are well written. However, as noted in the introduction with gen-AI the sentences are often long and complex. Authors should revisit the sections and add a personal touch. For instance, the heading Measures of Rigour for Qualitaive Research, the tone is quite formal, which can be a result of gen-AI. This could potentially just read as Measures of Rigor
Results
1. Consider exploring subthemes or specific case examples to further illuminate the complexities of the experiences under the theme "Emotional Fragmentation and Reproductive Self-Perception”. This section could benefit from a more in-depth analysis.
Discussion
1. Add a discussion on limitations
Author Response
- The introduction is quite long (unnecessarily). Consider tightening it by summarizing some of the literature review and focusing on the specific aims of the current study. Line 41 – 67 could be consolidated into one paragraph.
- We apologize. Following your suggestion the introduction has been shortened.
- More emphasis could be placed on the specific gap in knowledge that this research aims to address.
- Thank you for your comment. New text has been added emphasizing the knowledge gap this study addresses.
- There are some gen-AI phrasing patterns. I suspect authors used AI for language editing. Authors should disclose use of gen-AI if not already done so. The methods are well written. However, as noted in the introduction with gen-AI the sentences are often long and complex. Authors should revisit the sections and add a personal touch. For instance, the heading Measures of Rigour for Qualitaive Research, the tone is quite formal, which can be a result of gen-AI. This could potentially just read as Measures of Rigor
- DeepL has been used for language editing. We have disclosed so.
- Consider exploring subthemes or specific case examples to further illuminate the complexities of the experiences under the theme "Emotional Fragmentation and Reproductive Self-Perception”. This section could benefit from a more in-depth analysis.
- Thank you for your comment. Subthemes have been included in the structure of the findings section.
- Add a discussion on limitations
- A new paragraph has been included discussing the study limitations.
Reviewer 3 Report
Comments and Suggestions for Authors
Line 10: "Girls and women make up over half of those living with HIV... .". Do the authors refer to statistics or to a specific region? It is unclear.
Line 16: Please decipher WLWH
Line 18: Please decipher SDA (Appearing in the full form earlier in the text).
Line 19: "Twenty-six women living with HIV, including a transgender woman, were interviewed, with an average age of 39.3 years". Please reconsider the sentence. First, it sounds oddly on linguistic level, secondly it is not inclusive - why the authors put emphasis on "transgender woman" instead of mentioning "cisgender and transgender woman"?
Line 21: The authors state that "only 58% were employed". Which is the percentage of employed cis and transgender females from general population?
Introduction
Line 33: Living with HIV worldwide?
The introduction does not specify the geographic or demographic scope of the study population, which could strengthen the narrative. Is it due to a specific reason?
Moreover, it feels that the introduction could have better logical progression and be organized better. For instance, from discussing global statistics, the authors jump into discussing perinatal period, then challenges and perinatal period again.
Furthermore, specific challenges experienced by women living with HIV (e.g., mental health risks during the perinatal period, stigma, and psychosocial neglect) are mentioned multiple times, and could be streamlined.
Methodology
A brief textual summary of the key demographics (e.g., age range, health status) in the main body would enhance the reader’s understanding of the sample characteristics.
Moreover, the recruitment process could be more clearly explained (e.g., why was the sample size of 26 chosen?).
Also, the description of theoretical sampling is somewhat vague. Insights on how the authors decided on the characteristics or criteria used to select participants could strengthen the narrative.
Results
The authors provided pieces of information related to health, and financial status of the study participants. Yet, it would be beneficial to link them with the research topic. For instance, undetectable viral loads. Are they associated with higher confidence in pursuing motherhood, or does stigma or fear of transmission still influence decisions?
3.1 It would be beneficial to provide more insights on how emotional fragmentation manifests (e.g., through guilt, shame, grief).
3.4 This subsection could benefit from clearer transitions between the different emotional states (e.g., from fear to hope). Could the authors provide with more insights related to long-term emotional impacts of these emotional states?
All in all, although the authors integrate fragments of interviews, the results section seems to lack depth and synthesis. In fact, the relevant elements (e.g., fear, stigma, support, resilience) are described in isolation rather than showing how they interact with each other to shape women’s decisions about motherhood. A brief synthesis would strengthen the narrative.
Discussion
The authors discuss stigma in context of depression, anxiety, and self-stigma. How stigma is internalized and how it manifests in different social contexts (e.g., healthcare settings, family life, and broader society)? The authors provide a wide range of research, however there is no proper synthesis of these ideas. How specific types of support (e.g., counseling etc.) influence reproductive decisions over time? How can healthcare professionals improve their engagement with WLWH during reproductive decision-making?
Author Response
- Line 10: "Girls and women make up over half of those living with HIV... .". Do the authors refer to statistics or to a specific region? It is unclear
- We apologize. We have clarified this in the revised version.
- Line 18: Please decipher SDA (Appearing in the full form earlier in the text).
- All acronyms have been reviewed and explained
- Line 19: "Twenty-six women living with HIV, including a transgender woman, were interviewed, with an average age of 39.3 years". Please reconsider the sentence. First, it sounds oddly on linguistic level, secondly it is not inclusive - why the authors put emphasis on "transgender woman" instead of mentioning "cisgender and transgender woman"?
- Thanks for the comment. We have clarified this stating ‘women’
- Line 21: The authors state that "only 58% were employed". Which is the percentage of employed cis and transgender females from general population?
- After re-reading the paper and considering this data does not provide any insight to the global paper we have deleted this sentence.
- Line 33: Living with HIV worldwide?
- Thank you for you comment. We referred to globally and we have specified so in the new version.
- The introduction does not specify the geographic or demographic scope of the study population, which could strengthen the narrative. Is it due to a specific reason?
- Thank you for your comment. The focus of the study has been deliberately maintained on Spain to ensure alignment with the socio-cultural context, allowing for the collection of realistic and contextually relevant data. This approach enhances the study's rigor by grounding the findings within a specific and consistent cultural and social framework, as recommended in qualitative research to ensure credibility and transferability. We have clarified this in the revised version.
- Moreover, it feels that the introduction could have better logical progression and be organized better. For instance, from discussing global statistics, the authors jump into discussing perinatal period, then challenges and perinatal period again.
- Thank you. We have reformulated the introduction following your suggestion
- Furthermore, specific challenges experienced by women living with HIV (e.g., mental health risks during the perinatal period, stigma, and psychosocial neglect) are mentioned multiple times, and could be streamlined.
- Thank you for your comment. An effort has been made to streamline this same section in the discussions
- A brief textual summary of the key demographics (e.g., age range, health status) in the main body would enhance the reader’s understanding of the sample characteristics
- Sociodemographic data has been included in the results section.
- Moreover, the recruitment process could be more clearly explained (e.g., why was the sample size of 26 chosen.
- Thanks for your comment. We stated that the final sample size was unknown until the study progressed and theoretical data saturation was achieved
- Also, the description of theoretical sampling is somewhat vague. Insights on how the authors decided on the characteristics or criteria used to select participants could strengthen the narrative.
- We apologize. We have provided more details about theoretical sampling.
- The authors provided pieces of information related to health, and financial status of the study participants. Yet, it would be beneficial to link them with the research topic. For instance, undetectable viral loads. Are they associated with higher confidence in pursuing motherhood, or does stigma or fear of transmission still influence decisions
- Thank you for your comment. While the information on participants' health and financial status, such as undetectable viral loads, is valuable, it was not directly analyzed in relation to the research topic in this study. The inclusion of these characteristics primarily aimed to provide a clearer understanding of the sample, enhancing the transferability and credibility of the findings. Although it would indeed be insightful to explore whether undetectable viral loads are associated with higher confidence in pursuing motherhood, or whether stigma or fear of transmission still influence reproductive decisions, this aspect was not the focus of our analysis. Future research could delve into these questions to further understand how health factors impact reproductive decision-making in women living with HIV.
- It would be beneficial to provide more insights on how emotional fragmentation manifests (e.g., through guilt, shame, grief).
- We have included a sentence stating that participants also experienced guilt, shame, fear, and other related emotions.
- 3.4 This subsection could benefit from clearer transitions between the different emotional states (e.g., from fear to hope). Could the authors provide with more insights related to long-term emotional impacts of these emotional states?
- Thank you for your comment. The aim of the study was not identify or explain transitions. However, we have tried to present de findins using a more transitional style.
- All in all, although the authors integrate fragments of interviews, the results section seems to lack depth and synthesis. In fact, the relevant elements (e.g., fear, stigma, support, resilience) are described in isolation rather than showing how they interact with each other to shape women’s decisions about motherhood. A brief synthesis would strengthen the narrative.
- Thank you for your suggestion. Two new paragraphs have been added at the end the findings sections about the interplay of Factors in Women’s Reproductive Decisions with HIV to strengthen the narrative.
- The authors discuss stigma in context of depression, anxiety, and self-stigma. How stigma is internalized and how it manifests in different social contexts (e.g., healthcare settings, family life, and broader society)? The authors provide a wide range of research, however there is no proper synthesis of these ideas. How specific types of support (e.g., counseling etc.) influence reproductive decisions over time? How can healthcare professionals improve their engagement with WLWH during reproductive decision-making?
- We believe that with the new proposal for results and discussion, we have improved the presentation and discussion regarding stigma, anxiety, self-stigma, and depression.
Reviewer 4 Report
Comments and Suggestions for Authors
Thank you for the opportunity to review the manuscript. It is worth publishing, however, requires some editing before.
1. The methods part should be more detailed. Please ensure the following are covered: Study design, study population, study setting, exclusion and inclusion criteria, study instrument (questionnaire).
2. table 1 should be moved to the results part.
3. table 2 - the font size and resolution should be changed. I cannot see anything there. In the current view, it is unclear.
4. More visual materials are required to improve the comprehension of the study results. Please present it in figures.
5. Study strengths and limitations should be reported at the end of the discussion part.
6. The discussion should follow the outline suggested below:
Discussion
1.1 Rationale of the study (why it was done)
1.1.1 Main findings of the study
1.1.2 What makes your study unique
1.1.3 What it adds to what we already know
1.2 Study subjects
1.3 Subject of the discussion
Comparison of your results with previous studies in the field. Agreement and disagreement with the studies compared
1.4 Summ up of the study, study strengths and limitations
1.5 Clinical implication
Author Response
- The methods part should be more detailed. Please ensure the following are covered: Study design, study population, study setting, exclusion and inclusion criteria, study instrument (questionnaire).
- We apologize. We have rewritten the methods section following your recommendations.
- table 1 should be moved to the results part.
- We have deleted the table due to copyright issues and explained it in prose.
- table 2 - the font size and resolution should be changed. I cannot see anything there. In the current view, it is unclear.
- We apologize. We have provided a new version.
- More visual materials are required to improve the comprehension of the study results. Please present it in figures.
- A diagram has been included to facilitate comprehension.
- Study strengths and limitations should be reported at the end of the discussion part.
- The discussion has been rewritten.
Round 2
Reviewer 1 Report
Comments and Suggestions for Authors
The quality of the table is not acceptable. Table should be re-written, provide as table. The present version looks like Figure to me.
Figure 1 - some texts cannot be read. I think Sentence case text should be use not ALL CAPITALS.
Author Response
Thank you for you comment. We apologize for the poor quality of the table. We have deleted de image an pasted a real table. We have modified Figure 1 avoiding the use of capitals and using a big enough letter size. We hope it works now.
Reviewer 3 Report
Comments and Suggestions for Authors
The paper has been improved in a comprehensive way. A few comments: Please add separate practical implications, future directions and limitations section withing the discussion section. Please expand these elements, especially future directions. This will increase the scientific rigour and impact, as well as improve the readability.
Author Response
Thank you for your comment. We have added separate practical implications, future directions and limitations subsections at the end of the discussion section.